# Defining MRI Superiority over CT for Colorectal and Neuroendocrine Liver Metastases

**DOI:** 10.3390/cancers15205109

**Published:** 2023-10-23

**Authors:** Marc A. Attiyeh, Gautam K. Malhotra, Daneng Li, Saro B. Manoukian, Pejman M. Motarjem, Gagandeep Singh

**Affiliations:** 1Department of Surgical Oncology, Cedars-Sinai Medical Center, Los Angeles, CA 90048, USA; marc.attiyeh@cshs.org; 2Department of Surgery, USC, Keck School of Medicine, Los Angeles, CA 90033, USA; 3Department of Medical Oncology, City of Hope National Medical Center, Duarte, CA 91010, USA; 4Department of Radiology, City of Hope National Medical Center, Duarte, CA 91010, USA; 5Department of Surgery, City of Hope National Medical Center, Duarte, CA 91010, USA

**Keywords:** colorectal liver metastases, neuroendocrine liver metastases, imaging, staging, CT, MRI, surgery, Eovist

## Abstract

**Simple Summary:**

We compared CT and MRI for staging metastatic colorectal or neuroendocrine liver metastases (CRLMs and NELMs, respectively). Data from 76 patients (42 CRLMs; 34 NELMs) were analyzed, with two blinded radiologists (R1 and R2) independently assessing the images. For CRLMs, CT and MRI showed no significant differences in lesion number or size. However, in NELMs, Eovist^®^-enhanced MRI detected more lesions (*p* = 0.02) and smaller lesions (*p* = 0.03) than CT. In conclusion, CT and MRI are equivalent for CRLMs, but for NELMs, MRI outperforms CT, potentially impacting treatment planning and surgery.

**Abstract:**

Background: We compared CT and MRI for staging metastatic colorectal or neuroendocrine liver metastases (CRLMs and NELMs, respectively) to assess their impact on tumor burden. Methods: A prospectively maintained database was queried for patients who underwent both imaging modalities within 3 months, with two blinded radiologists (R1 and R2) independently assessing the images for liver lesions. To minimize recall bias, studies were grouped by modality, and were randomized and evaluated separately. Results: Our query yielded 76 patients (42 CRLMs; 34 NELMs) with low interrater variability (intraclass correlation coefficients: CT = 0.941, MRI = 0.975). For CRLMs, there were no significant differences in lesion number or size between CT and MRI. However, in NELMs, Eovist^®^-enhanced MRI detected more lesions (R1: 14.3 vs. 12.1, *p* = 0.02; R2: 14.4 vs. 12.4, *p* = 0.01) and smaller lesions (R1: 5.7 vs. 4.4, *p* = 0.03; R2: 4.8 vs. 2.9, *p* = 0.02) than CT. Conclusions: CT and MRI are equivalent for CRLMs, but for NELMs, MRI outperforms CT in detecting more and smaller lesions, potentially influencing treatment planning and surgery.

## 1. Introduction

Colorectal cancer (CRC) and neuroendocrine tumors (NETs) are among the most common malignancies that cause liver metastasis (LM). Although the diagnosis, prognosis, and treatment options vary, surgical resection can be potentially curative in both cases. CRC is the third most common malignancy worldwide, but death rates have decreased in the United States over the last 40 years [1,2]. Approximately 50% of patients with CRC develop liver metastases (CRLMs), of which 20% are amenable to potentially curative liver resection [1,3,4,5]. CRLMs can be synchronous, diagnosed at the time of primary diagnosis, or metachronous, detectable on imaging at a later time after primary diagnosis. NETs are tumors that arise from neuroendocrine cells located throughout the body [6]. They represent about 0.5% of all newly diagnosed malignancies and are relatively rare, with a worldwide incidence of approximately 2 per 100,000 people [7,8,9]. Neuroendocrine LMs (NELMs) are more common in midgut NETs (67–91%) than in pancreatic NETs (8–77%) [10].

Following the diagnosis of distant metastases in the liver, preoperative imaging is performed in patients with resectable disease to assess the location and extent of LMs, as well as determine the optimal surgical approach. The most common imaging modality used for the detection of LMs is contrast-enhanced computed tomography (CT), ideally with dedicated four-phase CT and thin slices (2.5 mm) [11,12]. In the absence of dedicated four-phase CT, the venous phase is often the most helpful modality for identifying liver lesions. Magnetic resonance imaging (MRI) has historically been shown to have better sensitivity and specificity than CT, especially with smaller lesions and when the liver shows signs of steatosis [13,14,15,16]. In one of the first published reports comparing CT and MRI for LMs, Reinig et al. reported that MRI found more detectable LMs than CT in a cohort of 20 patients (95.4% vs. 87.1%, respectively) [13]. A 2010 meta-analysis focusing only on CRLMs examined 39 articles (3391 patients from 1990 to 2010), which included a prospective design methodology, the use of both CT and MRI, and a histopathologic examination as the reference standard [15]. The sensitivity estimates for CT and MRI were 74.4% and 80.3%, respectively. A significant difference was observed in the smaller lesions (<10 mm). They also noted that the sensitivity of MRI increased in 2004, perhaps because of higher-resolution imaging, and concluded that MRI is more sensitive than CT and should be the preferred first-line modality for evaluating CRLMs. A 2021 meta-analysis of 12 prospective studies similarly found improved sensitivity with MRI compared to other imaging modalities [17]. Despite these findings, MRI has not been fully adopted as the imaging modality of choice for staging CRLMs. A nationwide population-based study in the Netherlands examined the trends and variations in the use of preoperative imaging modalities for CRLMs [18]. Of the 4510 patients who underwent a CT scan between 2014 and 2018, only 2855 (63%) underwent follow-up contrast-enhanced MRI. The authors noted that the trend is increasing, but the results highlight a large population of patients who may be underrepresented due to incomplete imaging.

While CRLMs are typically hypointense, NELMs exhibit early enhancement in the arterial phase, followed by a peripheral washout. In 2004, Dromain et al. compared the sensitivity of CT, MRI, and somatostatin receptor scintigraphy (SRS) for detecting NELMs, applied pathological correlation when available, and concluded that MRI was the more sensitive imaging modality [19]. In 2010, a small prospective study of 11 patients included a postoperative pathological examination to determine the “true” count of neuroendocrine liver metastases and compared it to preoperative imaging studies (ultrasound, CT, MRI, and somatostatin receptor scintigraphy (SRS)) [20]. None of the imaging modalities were able to detect greater than 50% of the lesions, with MRI demonstrating the highest mean accuracy (48.8%). A more recent study in 2015 compared CT, MRI, and Ga-68 DOTATATE in a relatively small cohort of 16 patients, and contrast-enhanced MRI performed the best, followed by noncontrast MRI when combined with diffusion-weighted imaging (DWI) [21]. Since these studies were conducted, there have been significant advances in MRI contrast agents (i.e., the use of Eovist^®^ (gadoxetate disodium) to improve hepatocyte uptake) and SRS (i.e., ^68^Ga-DOTATATE scans have effectively replaced octreotide scans). However, these studies underscored the importance of preoperative imaging in determining the optimal operative strategy to identify as many tumors as possible.

While the NCCN Clinical Practice Guidelines in Oncology (NCCN Guidelines^®^) state that hepatic MRI is “preferred” over CT to assess the number and distribution of CRLMs, no such recommendation is made in the NCCN Guidelines^®^ for NELMs [22,23,24]. The North American Neuroendocrine Tumor Society’s (NANETS’s) guidelines were recently updated to include the statement that hepatobiliary-phase liver MRI should be obtained in lieu of CT for the detection of liver metastases. However, this remains a grade C recommendation based on level 3 evidence [25]. The current retrospective observational study aimed to determine if MRI remains superior to CT for both CRLMs and NELMs and to test the hypothesis that MRI is superior to CT in detecting not only more lesions but also smaller ones, especially now that Eovist has emerged as a superior MRI contrast agent for NELMs [26,27].

## 2. Materials and Methods

A prospectively maintained database at the City of Hope National Medical Center was queried for patients who were diagnosed with either CRLM or NELM and who underwent both contrast-enhanced CT with a portal venous phase and MRI with or without contrast within three months. CT scans from PET/CT studies were not eligible for inclusion. Studies in which ablation had been previously performed were excluded. The data and imaging studies were anonymized and stored in a data repository separate from other clinical or radiological data. Two blinded radiologists (R1 and R2) with expertise in hepatic imaging (over 19 years of combined experience) independently evaluated the studies and recorded the numbers of lesions, sizes (in mm) of the smallest and largest lesions, presence of hepatic steatosis, and width of the CT slices. Cases with more than 20 lesions were counted as 20 at the time of analysis to prevent data skewing from outliers. When dealing with a multitude of metastases, we chose to set the lesion count at 20 to find a middle ground. This decision aimed to avoid setting the count too low, which might obscure potential differences between CT and MRI, and also prevent it from being set too high, which could impose an impractical burden on the radiologists. To minimize recall bias, the radiologists were first given a list of CT studies in a random order; two months later, they were given a list of MRI studies in a completely new random order.

Statistical analyses were conducted using Microsoft Excel, IBM SPSS Statistics (version 26), and R Statistical Software (v4.3.0; R Core Team 2023, Vienna, Austria) [28,29]. To determine the reliability of the results between the two raters, intraclass correlation coefficients (ICCs) and their 95% confidence intervals were calculated based on a mean rating (k = 2) and an absolute agreement two-way mixed-effects model. Fleiss’ kappa was used to determine the inter-rater agreement for the presence of steatosis on MRI. A paired two-sided Student’s t-test was performed on the number of lesions for CT and MRI for each radiologist. Results were considered statistically significant at a *p*-value of less than or equal to 0.05. A similar comparison was performed for the size of the smallest lesion, except that we limited our analysis to studies with at least two lesions to avoid skewing the results by large solitary lesions. A subgroup analysis was also conducted for MRIs, specifically with the more recently adopted Eovist contrast agent, to examine whether these studies performed better than their CT counterparts when focusing on detecting small lesions.

## 3. Results

A total of 108 patients met our initial search criteria for those with either CRLM or NELM who underwent both CT and MRI within 3 months of each other. We excluded 15 patients for whom imaging was not available. We further excluded 13 patients after a chart review revealed that their studies were greater than 3 months apart. Finally, four additional patients were excluded because the CT scans were performed without contrast. The remaining 76 patients (42 CRLM and 34 NELM) were included in the study group. For the overall cohort, the median number of days between CT and MRI was 40 (interquartile range (IQR): 22–56), and the median age at the time of the scans was 59 years (IQR: 51–65). In the CRLM cohort, there were 29 males (69%) and 13 females (31%); the NELM group included 17 males (50%) and 17 females (50%). Most patients in both cohorts underwent a CT scan prior to an MRI. Most MRI scans were performed with Eovist (*n* = 41 (54%)); the remainder were composed of scans using Gadovist, MultiHance^®^, Dotarem^®^, or an unknown contrast agent (Table 1). The overall interrater reliability between the two radiologists was excellent. The ICC for the number of lesions observed on CT was 0.941 (95% CI [0.907, 0.962]). Similarly, the ICC for the number of lesions observed on MRI was 0.975 (95% CI [0.961, 0.984]) (Table 2). There was some disagreement between the two radiologists regarding the presence of hepatic steatosis on MRI, as evidenced by the relatively low Fleiss’ kappa score (0.400).

In the CRLM cohort, there was no significant difference in the number of lesions observed on CT and MRI. The mean numbers of lesions and standard deviations seen on CT and MRI by radiologist 1 (R1) were 6.71 ± 5.88 and 7.81 ± 6.06 (*p* = 0.15), respectively. For radiologist 2 (R2), the mean numbers of lesions and standard deviations seen on CT and MRI were 7.24 ± 5.62 and 7.31 ± 5.62 (*p* = 0.89), respectively. Similarly, there was no difference in the size of the smallest lesions observed on CT and MRI. The mean sizes (in mm) of the smallest lesions seen on CT and MRI for R1 were 8.12 ± 8.85 and 8.10 ± 7.09 (*p* = 0.99). For R2, the means of the sizes of the smallest lesions seen on CT and MRI were 6.52 ± 5.81 and 5.40 ± 4.15 (*p* = 0.18). Restricting the analysis to MRI with Eovist or studies with two or more lesions did not yield statistically significant results (Table 3).

Conversely, in the NELM cohort, there was a significant difference in the number of lesions observed on CT and MRI. The means of the numbers of lesions seen on CT and MRI for the first radiologist (R1) were 12.09 ± 7.67 and 14.26 ± 7.43 (*p* = 0.02). For the second radiologist (R2), the means of the numbers of lesions seen on CT and MRI were 12.38 ± 6.91 and 14.38 ± 7.54 (*p* = 0.01), respectively (Figure 1 and Figure 2). When considering all cases, there was no difference between the sizes of the smallest lesions observed on CT and MRI. However, when we narrowed our analysis to MRIs with Eovist and when at least two lesions were seen on both CT and MRI, we identified a significant difference in the sizes of the smallest lesions between CT and MRI from both radiologists. For R1, the means of the sizes (in mm) of the smallest lesions seen on CT and MRI for R1 were 5.67 ± 1.85 and 4.43 ± 1.66 (*p* = 0.03), respectively. For R2, the mean sizes of the smallest lesions seen on CT and MRI were 4.77 ± 2.89 and 2.91 ± 1.41 (*p* = 0.02), respectively (Table 3).

## 4. Discussion

Currently, the use of CT and/or MRI for the staging of colorectal and neuroendocrine liver metastases has not been standardized. Existing guidelines stop short of making definitive recommendations for the use of MRI over CT for CRLMs and NELMs. However, as previously discussed, obtaining optimal hepatic imaging has the potential to determine resectability and surgical approaches and can even affect postoperative recurrence rates [30]. In this study, we demonstrated that two independent radiologists detected more and smaller NELMs on MRI than on conventional contrast-enhanced CT. We did not observe similar results in the CRLM cohort, with both imaging modalities appearing equivalent. One of the strengths of our study was that the comparisons were performed between scans for the same patient; thus, the typical confounding patient factors that often occur in observational studies are not applicable here.

Eovist (gadoxetate disodium) has shown superior diagnostic performance for detecting and characterizing small liver lesions, particularly in patients undergoing curative liver surgery (Figure 3). Additionally, Eovist-enhanced liver MRI has demonstrated potential benefits in the follow-up assessment of hepatic metastases, providing valuable information in challenging cases where other imaging modalities may be limited [31]. A recent 2018 study analyzed the MRIs of 30 patients with NELMs to determine the optimal phase and contrast agent for assessing tumor detectability, lesion size, and tumor–liver interface (TLI). The authors concluded that the hepatocellular phase with Eovist was the optimal choice [26]. The results of our study were consistent with this finding, as evidenced by the analysis of the smallest lesions. Initially, we observed no difference in the size of the smallest lesions in either modality or tumor type. However, when we narrowed our analysis to MRI scans performed with Eovist and cases where at least two lesions were noted on both CT and MRI, we uncovered a significant improvement in the detection of small NELMs on MRI by both radiologists. Like our previous results, we did not observe a similar effect in the CRLM group. This discrepancy can be primarily attributed to the inherent differences in the histological characteristics and vascularity of the two types of metastases. NELMs often have a rich blood supply and are typically hypervascular, leading to a hyperintense appearance on MRI. In contrast, CRLMs may exhibit a more variable appearance depending on factors such as size, vascularity, and degree of fibrosis.

Hepatic steatosis can lead to alterations in liver signal intensity on MRI. This is due to the presence of fat, which can potentially affect the visualization of lesions and their distinguishing features. Some of the key MRI features for diagnosing hepatic steatosis include hyperintensity on T1-weighted imaging, hypointensity on T2-weighted imaging, and opposed-phase imaging where there is a loss of signal intensity on the opposed-phase images compared to in-phase images, known as “chemical shift artifact”. In our study, we noted a relative disagreement between radiologists concerning the diagnosis of hepatic steatosis on MRI. The Fleiss’ kappa score was 0.400, which is generally interpreted as “fair” agreement (Table 2). This was not as strong as the previously mentioned ICC score regarding the numbers and sizes of hepatic lesions [32]. This result can be due to the complexity of diagnosing hepatic steatosis and the importance of combining imaging findings with patient history, clinical symptoms, laboratory tests, and additional imaging modalities, such as ultrasound. Nevertheless, the difference in MRI evaluation only serves to strengthen our primary finding of MRI superiority over CT in detecting NELMs, since two radiologists, who may interpret MRI findings differently, consistently agreed on finding more and smaller NELMs than when using CT.

We acknowledge that our findings diverge from previous studies examining the utility of CT and MRI in colorectal liver metastases (CRLMs). We propose several reasons for this variance between our data and previously published results. First, the continuous advancements in CT imaging technology have substantially improved image resolution and quality, consequently reducing the diagnostic gap between MRI and CT in detecting CRLMs. Second, the expertise of our radiologists may have played a pivotal role in their capacity to identify more lesions on CT scans, potentially diminishing the perceived significance of MRI in these cases. Third, the absence of a statistically significant finding in our study might be attributed to our sample size, as it is conceivable that the effect size in CRLMs is relatively small, necessitating a larger sample size for detection.

Although the present study offers valuable insights, it is important to interpret its findings while considering several inherent limitations. First, one potential origin of selection bias could be the order in which the medical imaging tests were requested, with MRIs sometimes ordered after CT scans, potentially due to suboptimal CT results. However, it is worth noting that within the NELM cohort, one-third of cases had an MRI as the initial diagnostic test. Second, we did not establish a pathological gold standard for the quantification of liver burden; therefore, we were unable to draw any conclusions regarding the sensitivity and specificity of either imaging modality. The diagnostic accuracy of the lesions observed on imaging was purely based on the experience of the two radiologists. Although the validity of the results is strengthened by the excellent concordance between the two radiologists, future work should include pathological correlations to reinforce our findings. In addition, the variation in CT protocol and scan slice size could have influenced the observed impact of MRI on the NELM population. Notably, 21% of the CT scans had a slice size of 5 mm, in contrast to 14% in the CRLM group. These limitations could potentially explain the equivalent result in the CRLM group, which contrasts with previously published studies [15,17]. Also, within the NELM cohort, 10 CT scans did not include an arterial phase, potentially leading to an overestimation of the effectiveness of MRIs in detecting these lesions. To conduct a consistent analysis and mitigate this potential confounding factor, future studies should establish standardized imaging protocols. For example, they should only use thin (1-3 mm) CT/MRI slices and always include an arterial phase, both in CT and MRI contrast-enhanced studies.

## 5. Conclusions

In conclusion, for patients with NELMs, MRI is superior to CT in detecting more lesions. We also conclude that the use of Eovist can help detect smaller NELMs on MRI when two or more lesions are present. Further studies will need to expand this analysis to include pathological correlations (either by biopsy or resection). Furthermore, implementing standardized imaging timing and protocols for both CT and MRI has the potential to eliminate confounding factors and enhance the robustness of the findings.

## Figures and Tables

**Figure 1 cancers-15-05109-f001:**
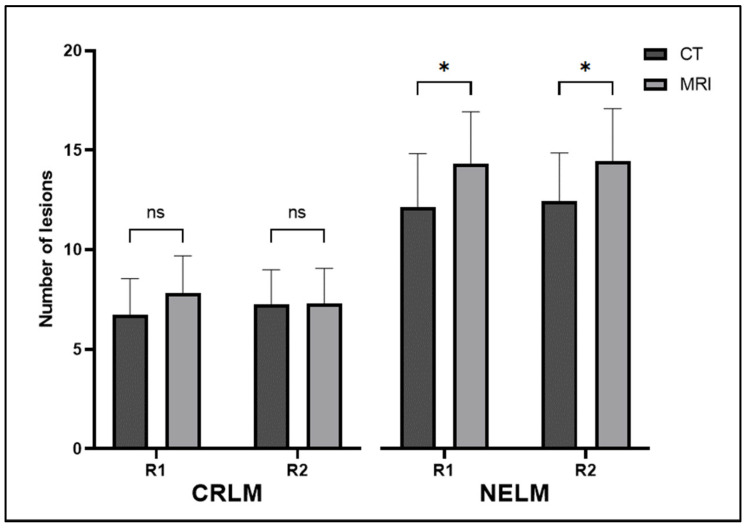
Mean numbers of lesions seen on CT and MRI by both radiologists for both CRLM and NELM. Error bars represent 95% confidence intervals. ns—not significant; *—significant; R1—radiologist 1; R2—radiologist 2; CRLM—colorectal liver metastasis; NELM—neuroendocrine liver metastasis.

**Figure 2 cancers-15-05109-f002:**
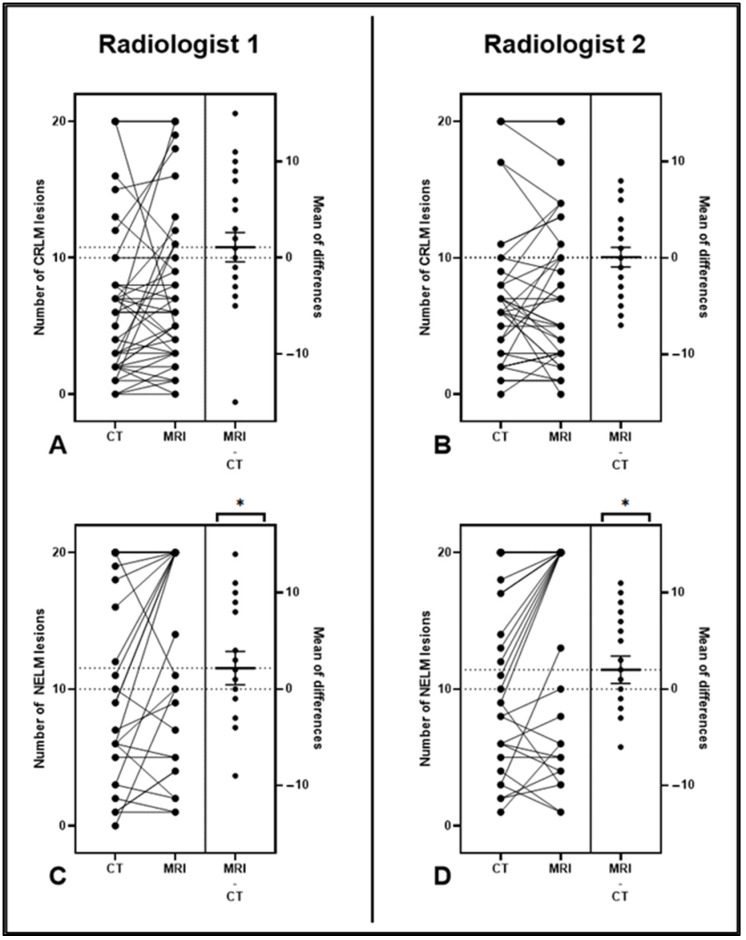
(**A**–**D**) Estimation plots demonstrating the magnitude of the differences between CT and MRI studies for CRLM (**A**,**B**) and NELM (**C**,**D**). The left axis displays a connected pair of data points for each patient, representing the number of lesions observed in their corresponding imaging studies. The right side shows the effect size (the number of MRI lesions minus the number of CT lesions) with a mean and 95% confidence intervals (CIs). Note that the 95% CI bars did not cross the zero line for the NELM cohort (**C**,**D**), demonstrating that a significantly higher number of lesions was seen on MRI. CRLM—colorectal liver metastasis; NELM—neuroendocrine liver metastasis; *—significant.

**Figure 3 cancers-15-05109-f003:**
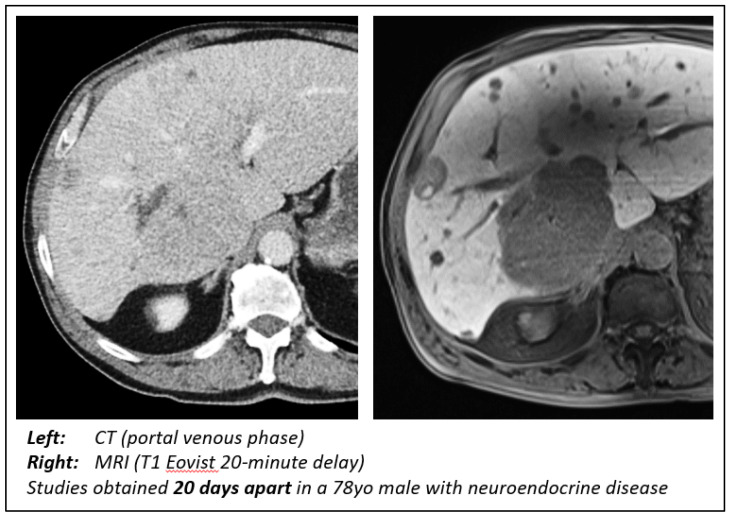
Comparison of CT and MRI findings. Note the ability of MRI with Eovist to not only better delineate the large metastasis but also illustrate disease burden throughout the liver.

**Table 1 cancers-15-05109-t001:** Patient characteristics from the CRLM and NELM cohorts. Median (IQR) or *N* (%).

Variable	CRLM (*n* = 42)	NELM (*n* = 34)
**Age**	54 (49–61)	60 (55–66)
**Gender**		
**Male**	29 (69%)	17 (50%)
**Female**	13 (31%)	17 (50%)
**Year of studies**		
**<2018**	3 (7%)	5 (15%)
**2018–2019**	30 (71%)	17 (50%)
**2020–2021**	9 (21%)	12 (35%)
**Days between CT and MRI scans**	32 (20–52)	46 (24–67)
**Scans chronological order**		
**CT first**	36 (86%)	23 (68%)
**MRI first**	6 (14%)	11 (32%)
**CT scan slice size (mm)**		
**1.25**	1 (2%)	0 (0%)
**2.50**	24 (57%)	24 (71%)
**3.00**	11 (26%)	3 (9%)
**5.00**	6 (14%)	7 (21%)
**MRI contrast agent**		
**Eovist**	31 (74%)	24 (71%)
**MultiHance**	3 (7%)	6 (18%)
**Gadavist**	4 (10%)	1 (3%)
**Dotarem**	1 (2%)	0 (0%)
**Unknown**	3 (7%)	3 (9%)
**Arterial phase available**		
**CT**	8 (19%)	24 (71%)
**MRI**	40 (95%)	33 (97%)

IQR—interquartile range; CRLM—colorectal liver metastasis; NELM—neuroendocrine liver metastasis; Eovist—gadoxetate disodium; MultiHance—gadobenate dimeglumine; Gadavist—gadobutrol; Dotarem—gadoterate meglumine.

**Table 2 cancers-15-05109-t002:** Statistical analysis of the interrater agreement. Median (IQR), *N* (%), or [95% CI].

Variable	CT (*n* = 76)			MRI (*n* = 76)		
	R1	R2	Statistics	R1	R2	Statistics
**Number of lesions**	7 (3–17)	8 (4–17)	ICC = 0.941 ^a^ [0.907, 0.962]	9 (4–20)	9 (3–20)	ICC = 0.975 ^a^ [0.961, 0.984]
**Steatosis ^c^**						
**Present**	N/A	N/A	N/A	43 (57%)	23 (30%)	κ = 0.400 ^b^ [0.227, 0.573]
**Absent**	N/A	N/A		33 (43%)	53 (70%)	

^a^ ICC—intraclass correlation coefficient, a measure of reliability between the two raters (≤0.50: poor; 0.50 < *x* ≤ 0.75: moderate; 0.75 < *x* < 0.90: good; ≥0.90: excellent). ^b^ Fleiss’ kappa, a measure of interrater agreement (<0.40: poor; 0.40 ≤ *x* ≤ 0.75: fair to good; >0.75: excellent). ^c^ Steatosis was only assessed on an MRI. IQR—interquartile range; CI—confidence interval; R1—radiologist 1; R2—radiologist 2.

**Table 3 cancers-15-05109-t003:** Comparison of CT and MRI. Median (IQR).

Variable		CRLM (*n* = 42)			NELM (*n* = 34)		
		CT	MRI	*p*-Value	CT	MRI	*p*-Value
**Number of lesions**	R1	6 (2–8)	6 (3–11)	0.15	12 (6–20)	20 (8–20)	0.02 ^†^
	R2	6 (3–9)	5 (3–11)	0.89	13 (6–20)	20 (6–20)	0.01 ^†^
**Number of lesions with steatosis ^a^**	R1	4 (2–7)	5 (3–11)	0.20	10 (3–20)	13 (6–20)	0.09
	R2	6 (4–7)	4 (2–9)	0.61	9 (3–17)	20 (6–20)	0.01 ^†^
**Smallest lesion (mm)**	R1	6 (5–10)	8 (4–11)	0.99	6 (5–8)	5 (4–6)	0.36
	R2	4 (3–9)	4 (3–6)	0.18	4 (3–5)	3 (2–4)	0.83
**Smallest lesion (mm) with *n* ≥ 2 ^b^**	R1	6 (5–11)	8 (5–11)	0.78	5 (5–7)	4 (3–5)	0.39
	R2	4 (3–8)	4 (3–6)	0.15	4 (3–5)	3 (2–4)	0.01 ^†^
**Smallest lesion (mm) with Eovist, *n* ≥ 2 ^c^**	R1	6 (5–10)	8 (5–9)	0.72	5 (5–6)	4 (3–5)	0.03 ^†^
	R2	4 (3–7)	4 (3–5)	0.14	4 (3–5)	2 (2–3)	0.02 ^†^

^†^ Statistically significant (*p* < 0.05). ^a^ Limited to cases where hepatic steatosis was noted on the MRI by the respective radiologist. ^b^ Limited to cases where greater than or equal to two lesions were seen on both CT and MRI. ^c^ Limited to cases where the MRI was performed with Eovist and where greater than or equal to two lesions were seen on both CT and MRI. CRLM—colorectal liver metastasis; NELM—neuroendocrine liver metastasis; Eovist—gadoxetate disodium; IQR—interquartile range; R1—radiologist 1; R2—radiologist 2.

## Data Availability

Data is unavailable due to privacy issues.

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
