# Peer review of "Defining MRI Superiority over CT for Colorectal and Neuroendocrine Liver Metastases"

_cancers, 2023, doi:10.3390/cancers15205109_

Round 1

Reviewer 1 Report

1- More recent references are necessary in the literature.

2- Authors need to provide some information about the type of study.l

3- In Discussion section, there is no comparison between the results of this study with the previous studies in this field.

Minor English language editing is needed.

Reviewer 2 Report

General comments:

I believe the Authors should report this study as a personal experience. The selected series is very heterogeneous.

 Instead the review of  CT, MRI and PET  clinica value and accuracy in detecting NELMs and CRCLs is well conducted and interesting. I would focus the review on this analysis, by reducing and shortening the value of the study which may be interesting but it is unfortunately affected by several bias.

The study design  is too weak to produce reliable results.  For example, in Materials and methods the Authors mention: "CRLMs or NELMs underwent contrast-enhanced CT with portal venous phase and MRI with or without contrast within three months"

 Selection criteria are not sufficient to compare the two series, because:

1) A venous phase of CT can hardly detect NELMS (hypervascular lesions); an arterial phase should always be included, at least for NELMs, otherwise these lesions should not be considered for a comparative study WITH MRI. 

2) Authors should at least report in how many cases of NELMs the arterial phase was not available. The lack of arterial phase may largely explain the different results between CRLM and NELM.

3) Also, the term "MRI with or without contrast" is too generic. Please specify if the contrast enhnced phase included an arterial phase or not in MRI as well. Also please specify if all patients performed a contrast enhancement phases, and how many acquisitions during the contrast injection (arterial? venous? delayed? epatospecific?) 

4) The  pre-contrast MRI protocol is not specified: for example, dit it include DWI? T2 weighted fat suppressed images? etc

Indeed, as expected, there was a significant difference in the number of lesions observed on CT and MRI in the NELM cohort. " The averages of the number of lesions observed at CT and MRI for the first radiologist (R1) were 12.09±7.67 and 14.26±7.43 (p = 0.015). For the second radiologist (R2), the averages of the number of lesions observed at CT and MRI were 12.38±6.91 161 and 14.38±7.54, respectively (p = 0.007)."

In addition: in 21% of CT scans, the slice size was 5 mm (too high!!), in contrast to 14% in the CRLM group.

The Authors should better emphasize  the limitations of the study, for example  by adding in this sentence, "To conduct a consistent analysis and mitigate this potential confounding factor, future studies should establish standardized imaging protocols, using  only thin (1-3 mm) CT/MRI slices and always including the arterial phase, both  in CT and MRI contrast enhanced studies ."

Reviewer 3 Report

1 - There is no ethical statement about the handling of the patient data. Approval /permission to use the data by a review board?

2 - The two radiologists R1 and R2 have experience with hepatic imaging. Please specify how many years of experience.

3 - Data is obtained from a database by query for patients who were diagnosed with either CRLM or NELM and who underwent both contrast-enhanced CT with a portal venous phase and MRI with or without contrast within three months. Not described is any reason why a second scan is made within three month? If the quality of the CT scan is not conclusive another scan is likely to be requested. Selection bias may influence the results.

4 - Data seems to be quite heterogeneous, with respect to CT slice thicknesses do vary 1.25, 2.5, 3 and even 5 mm are stated. There is no referral to the vendors of the CT, the X-ray factors (kV, mAs) collimation are not specified, reconstruction method remains unknown. Yet these factors do influence the quality of the CT scan and its discrimination power regarding the diagnosis. The amount of contrast injected, the speed of injection and the CT trigger also play a role. Regarding the MRI data, there is no vendor, pulse sequence nor the magnetic fieldstrength in Tesla is stated. From the caption of figure 3 one can read that apparently a T1 scan is read. 

5 - The reading by the radiologists is not very well described, viewing conditions as normal reading? How about the time of reading, at the end of the normal dayload shift or during lunchbreak? Both radiologist the same random sequence list? 

6 - The statistical analysis is described as executed, hardy any motivation for the choices of the tests are provided. The is no power analysis provided for the type of differences that can be found.

7 - There are two radiologists involved, interrater differences are investigated. However, no intra observer variability. Adding more readers and investigating also intraobserver variability will provided a scientifically stronger paper.

8 - There is no groundtruth about the radiologist readings (of course a radiologist is always certain, however, not always right) by pathology. That this must be done is stated in the conclusion but the authors of the paper made the choice not to do it.

9 - Is the found higher detectability of Iovist MRI clinically relevant,  will it lead to a different staging outcome and thus a possibly other treatment of choice? 

10 - The final line of the conclusion " .... implementing standardized imaging timings and protocols for both CT and MRI has the potential to eliminate confounding factors and enhance the robustness of the findings" is very important to reduce the variability of the data. This is a just statement.

Reviewer 4 Report

The objective of this study is to assess the efficacy of MRI compared to CT in diagnosing and staging colorectal liver metastases (CRLM) and neuroendocrine liver metastases (NELM), with a unique focus on incorporating preoperative imaging data for lesion detection. Several key points need clarification in this study:

1. The author references existing literature and NCCN Guidelines, which suggest that MRI is superior to CT in evaluating CRLMs. However, the study's findings indicate that there are no significant differences between CRLM detection in CT and MRI. This apparent deviation from established practices warrants explanation.

2. The study does not specify the particular MRI imaging modality used or whether contrast agents were employed, which can significantly affect diagnostic accuracy. This lack of detail can compromise the study's reproducibility and fairness when comparing it to contrast-enhanced CT imaging.

3. The absence of a gold standard for lesion assessment, with comparisons solely relying on the interpretations of two radiologists, raises concerns about the study's robustness. While this might be the available data, caution should be exercised not to overstate MRI's superiority over CT.

4. It's worth discussing why a limit of 20 lesions was set and the significance of assessing the smallest lesions in this research. These factors should be elucidated to provide a clearer context for the study's design.

In summary, this research presents an interesting work but it requires refinements in its research design to strengthen its credibility and convince readers of its robustness.

Round 2

Reviewer 3 Report

@4 You still do not specify your data, nor what type of images the radiologists have been reading. Scientific conclusions are rather weak in that regard.

@6 You did not pick up the suggestion for a power analysis. In case of NELM you have N = 34 patients, yet you state p - values in 3 decimals, how can these decimals be significant?  (abstact and page 5). Please do check guidelines on reporting about p - values.

@9 Perhaps, however, you have not convinced me and I do not share your two beliefs. Without ground truth, setting up a RCT would be a better way to find the distinction which is the basic research question of the current paper.

Reviewer 4 Report

My comments have been addressed.

Author Response

Dear Reviewer,

Thank you for your prompt review and for expressing satisfaction with our responses. Your valuable feedback has greatly contributed to the improvement of our paper.

Round 3

Reviewer 3 Report

no futher comments